# Deep Insights into Gut Microbiota in Four Carnivorous Coral Reef Fishes from the South China Sea

**DOI:** 10.3390/microorganisms8030426

**Published:** 2020-03-18

**Authors:** Yu-Miao Gao, Ke-Shu Zou, Lei Zhou, Xian-De Huang, Yi-Yang Li, Xiang-Yang Gao, Xiao Chen, Xiao-Yong Zhang

**Affiliations:** 1Joint Laboratory of Guangdong Province and Hong Kong Region on Marine Bioresource Conservation and Exploitation, College of Marine Sciences, South China Agricultural University, 483 Wushan Road, Guangzhou 510642, China; gym_Seven@163.com (Y.-M.G.); zoukeshu@scau.edu.cn (K.-S.Z.); zhoulei@scau.edu.cn (L.Z.); huangxd@scau.edu.cn (X.-D.H.); liyiyang0731@126.com (Y.-Y.L.); 2Guangdong Laboratory for Lingnan Modern Agriculture, Guangzhou 510642, China; 3Guangdong Provincial Key Laboratory of Nutraceuticals and Functional Foods, College of Food Science, South China Agricultural University, Guangzhou 510642, China; gaoxiangyang@scau.edu.cn

**Keywords:** gut microbiota, carnivorous coral reef fishes, Illumina sequencing, the South China Sea

## Abstract

Investigations of gut microbial diversity among fish to provide baseline data for wild marine fish, especially the carnivorous coral reef fishes of the South China Sea, are lacking. The present study investigated the gut microbiota of four carnivorous coral reef fishes, including *Oxycheilinus unifasciatus*, *Cephalopholis urodeta*, *Lutjanus kasmira*, and *Gnathodentex aurolineatus*, from the South China Sea for the first time using high-throughput Illumina sequencing. Proteobacteria, Firmicutes, and Bacteroidetes constituted 98% of the gut microbiota of the four fishes, and 20 of the gut microbial genera recovered in this study represent new reports from marine fishes. Comparative analysis indicated that the four fishes shared a similar microbial community, suggesting that diet type (carnivorous) might play a more important role in shaping the gut microbiota of coral reef fishes than the species of fish. Furthermore, the genera *Psychrobacter*, *Escherichia-Shigella*, and *Vibrio* constituted the core microbial community of the four fishes, accounting for 61–91% of the total sequences in each fish. The lack of the genus *Epulopiscium* in the four fishes was in sharp contrast to what has been found in coral reef fishes from the Red Sea, in which *Epulopiscium* was shown to be the most dominant gut microbial genus in seven herbivorous coral reef fishes. In addition, while unique gut microbial genera accounted for a small proportion (8–13%) of the total sequences, many such genera were distributed in each coral reef fish species, including several genera (*Endozoicomonas*, *Clostridium*, and *Staphylococcus*) that are frequently found in marine fishes and 11 new reports of gut microbes in marine fishes. The present study expands our knowledge of the diversity and specificity of gut microbes associated with coral reef fishes.

## 1. Introduction

Coral reefs, the most diverse and productive marine ecosystems on earth, are recognized as the rainforests of the sea [1]. It has been reported that coral reefs play a very important role in maintaining biodiversity and ecological balance of the oceans, although they occupy less than 0.1% of the world’s ocean surface and provide complex and manifold marine habitats that support 25% of all marine species [2]. However, under the influence of climate change, disease outbreaks, overfishing, and even bad weather like hurricanes [3], the corals worldwide are undergoing great degradation, which also threatens the diversity of coral reef fishes [4]. As one of the most significant components of coral ecosystems, coral reef fishes can take a crucial role in protecting coral reef ecosystems from pathogen invasion. Most of the recent studies have mainly focused on investigating the diversity and distribution of coral reef fishes. So, despite the unparalleled species diversity and population density of coral reef fish reported in recent decades [5], very little is known about their intestinal tract microbiology [6].

The fish-associated symbiotic gut microbiota play a crucial role in nutritional provision and metabolic homeostasis. Nikouli et al. found that the gut microbial communities of farmed sea bream (*Sparus aurata*) fed different diets could influence the host’s nutritional intake [7]. Another study reported that the intestinal microbiota of grass carp (*Ctenopharyngodon idellus*) fed with faba beans could significantly enhance the host’s metabolic functions [8]. The gut microbiota can prevent the colonization of some infectious agents and maintain the host’s immunity. Gupta et al. discovered that antibiotic-induced perturbations in Atlantic salmon could be regulated by the main gut microbial community of salmon [9]. Recent studies on the gut microbiota of fish have mainly focused on the manipulation of their diets and gut microbial communities to meet the needs of fish farming while trying to maintain host health and welfare.

However, investigations of gut microbial diversity among fish to provide baseline data from wild marine fish, especially from coral reef fishes, are still lacking. A few studies have revealed that the gut microbial community of the coral reef fish *Kyphosus sydneyanus* is markedly more abundant in larger fish than in smaller fish and that there is no specific microbial community distributed between the foregut and hindgut or between large fish and small fish [10]; the diet has a strong influence on the gut microflora, and there is a statistically significant correlation between the host phylogeny and gut microbiota in surgeon fishes [11]. Only a few studies on the gut microbiota of coral reef fishes have focused on herbivorous species and fermentation. The microbial community composition of carnivorous coral reef species has been relatively poorly investigated. Carnivorous fish exhibit a relatively high trophic status in the food chain and play a crucial role in maintaining the ecological balance of coral reef systems.

The goal of this study was to comprehensively investigate the diversity and structure of the microbial community associated with four carnivorous coral reef fishes, *Oxycheilinus unifasciatus*, *Cephalopholis urodeta*, *Lutjanus kasmira*, and *Gnathodentex aurolineatus*, from the South China Sea using high-throughput Illumina sequencing. Furthermore, we compared the gut microbial distribution in the four fish species and defined the core and unique microbial community, which provided data on the dominant gut microbiota in coral reef fishes from the South China Sea. Such information will expand our knowledge about the diversity and specificity of microbes associated with carnivorous fish species in coral reefs and help us understand the role of specific gut microbes. Based on the present study, we should achieve a better understanding of the relationship between the dietary and trophic structure of coral reef fish and their gut microbiota.

## 2. Materials and Methods

### 2.1. Sample Collection

Four species of coral reef fishes and 12 samples (Samples Ou1~Ou3; Cu1~Cu3; Lk1~Lk3, and Ga1~Ga3) were collected from Zhubi Reef (10°55′ N, 114°03′ E) in the South China Sea in Aug 2017. All the coral reef fishes were identified by Dr. Xiao Chen (South China Agricultural University, Guangzhou, China) as belonging to the four species of *Oxycheilinus unifasciatus* (Samples Ou1~Ou3), *Cephalopholis urodeta* (Samples Cu1~Cu3), *Lutjanus kasmira* (Samples Lk1~Lk3), and *Gnathodentex aurolineatus* (Samples Ga1~Ga3) and were categorized as carnivorous coral reef fishes according to their diet in reference to descriptions from FishBase (https://www.fishbase.in/search.php, Copenhagen, Denmark). In addition, the phylogenetic tree based on the sequences of the cytochrome c oxidase subunit I (COI) genes of the four fish species downloaded from National Center for Biotechnology Information (NCBI, Bethesda, MD, USA) represented the evolutionary relationships between the four fishes. *O. unifasciatus*, *G. aurolineatus*, and *L. kasmira* were clustered onto one branch, which indicated that *C. urodeta* was less closely related to the other three fishes (Figure 1).

These adult coral reef fishes were collected using hook and line and kept in sterile sea water. After anesthetization with 60 mg/L tricaine methanesulfonate, each fish was transported on ice to the laboratory as quickly as possible [11]. Under aseptic conditions, the external surface of the fish was cleaned with 75% ethanol to avoid contamination by the surface microbes on the fish. After opening the ventral surface using sterile scissors, the whole gut of each fish was aseptically removed and collected, and at least three replicates per sample were taken and assembled as one gut sample. A total of 12 gut samples from the four coral reef fishes was collected in frozen tubes and stored at −80 °C until DNA extraction.

### 2.2. Microbial DNA Extraction and 16S rRNA Amplicon Sequencing

The microbial DNA of each gut sample was extracted using the EZNA stool DNA Kit (Omega Biotek, Norcross, GA, USA) according to the manufacturer’s protocols [1]. The V3-V4 region of the microbial 16S rRNA gene was amplified by PCR (initial denaturation at 95 °C for 2 min, followed by 27 cycles at 98 °C for 10 s, 62 °C for 30 s, and 68 °C for 30 s, with a final extension at 68 °C for 10 min) using the forward primer CCTACGGRRBGCASCAGKVRVGAAG and the reverse primer GGACTACNVGGGTWTCTAATTCC, where the barcode was an eight-base sequence unique to each sample. PCR amplification was performed in triplicate in 50 μL reactions containing 5 μL of 2.5 mM dNTPs, 5 μL of 10× KOD buffer, 1.5 μL of 5 μM forward and reverse primers, 1 μL of KOD polymerase, and approximately 100 ng of template DNA.

The microbial 16S rRNA amplicon of each sample of the four fish species was extracted and purified with the AxyPrep DNA Gel Extraction Kit (Axygen Biosciences, Union City, CA, USA) following the manufacturer’s operation guide. The purified microbial 16S rRNA amplicons were pooled in equimolar quantities and subjected to paired-end sequencing on the Illumina MiSeq platform (Illumina, San Diego, CA, USA) according to the manufacturer’s protocols [12].

### 2.3. Microbial Operational Taxonomic Units (OTUs) Cluster and Taxonomic Annotation

The raw reads were merged as raw tags with FLAST (v 1.2.11) according to an overlap of more than 10 bps and were further analyzed with Quantitative Insights Into Microbial Ecology (QIIME) software (Knight Lab, San Diego, CA, USA) (Version 1.9.1) [13,14], with processing under specific filtering conditions to improve the acuity of rare operational taxonomic unit (OTU) discrimination. The tags were searched against the reference database UCHIME (v 4.2) to remove chimeric sequences, and the effective tags were obtained. The further analyzed effective tags were clustered into operational taxonomic units (OTUs) at a 97% similarity level using VSEARCH (1.9.6). In addition, the effective tag with the highest abundance was selected as a representative sequence for each cluster, which was assigned taxonomically using SILVA release 128 (Max Planck Institute for Marine Microbiology and Jacobs University, Bremen, Germany) [15].

### 2.4. Alpha Diversity and Statistical Analysis

Based on the OTU analysis results, a random sampling method was used to generate a rarefaction curve, and alpha diversity analysis was performed in R (R Core Team, Vienna, Austria) [16]. Chao 1 and the Shannon index were calculated and used throughout the study as indicators of richness and diversity, and other indices were included in tables for easy comparison with the existing literature [14]. Venn diagram analysis identifying core and unique OTUs was performed in R (R Core Team, Vienna, Austria) between groups [16]. The Ribosomal Database Project (RDP) classifier Bayesian algorithm was employed to carry out species taxonomy analysis on the representative sequences of OTUs and to quantify the community composition of each sample at different classification levels. The Unweighted Pair Group Method with Arithmetic Mean (UPGMA) phylogenetic tree was constructed by unweighted group averaging via hierarchical clustering [17].

### 2.5. Nucleotide Sequence Accession Numbers

All sequence data from the 12 coral reef fish samples were deposited in the Sequence Read Archive of the NCBI under accession number PRJNA603639.

### 2.6. Ethics Statement

All experiments were carried out in strict accordance with the recommendations of the Guide for the Care and Use of Laboratory Animals of the National Institutes of Health. The use of animals in this study was approved by Animal Ethics Committee at the South China Agricultural University, China (approval ID: 201004152, January 2010).

## 3. Results

### 3.1. Sequence Overview

After denoising and chimaera detection, a total of 677,841 microbial 16S V4 sequences with a mean length of 458 bp were obtained and used for analysis, which were clustered into 191 OTUs with 97% sequence similarity. The average numbers of microbial OTUs detected in *O. unifasciatus*, *C. urodeta*, *L. kasmira*, and *G. aurolineatus* were 106, 125, 80, and 72, respectively (Table 1). Rarefaction curves for the four fishes constructed for the number of observed microbial sequences vs. OTUs showed a plateau ranging from 40 to 70 (Figure 2), indicating that the number of the obtained sequences could sufficiently represent the diversity of the gut microbes in each fish species. These results were supported by the comparison of the Shannon and Chao 1 indices and observed OTUs at 97% sequence similarity (Table 1). Good’s nonparametric coverage estimate indicated that more than 99.9% of the diversity was recovered in each fish sample.

### 3.2. Alpha Diversity

The variation of the gut microbiota within each fish species was reflected in the alpha-diversity estimates obtained at 97% sequence identity (Table 1). *C. urodeta* exhibited the highest expected Chao 1 richness values and the highest variability between replicates (*n* = 3; 67 ± 31). *O. unifasciatus* also presented high variability between replicates (*n* = 3; 54 ± 30). In contrast, *G. aurolineatus* displayed the lowest expected mean richness values and the lowest variability in richness values between replicates (*n* = 3; 46 ± 4). *L. kasmira* also showed low variability in the richness values between replicates (*n* = 3; 53 ± 9). There was similar inter- and intraspecies variability in the Shannon index (Table 1). *C. urodeta* (*n* = 3; 2.97 ± 0.40) exhibited the lowest variation with the highest average diversity. *O. unifasciatus* (*n* = 3; 2.89 ± 1.06) presented the highest variation with comparably high diversity. The average diversity of *L. kasmira* (*n* = 3; 2.55 ± 0.46) and *G. aurolineatus* (*n* = 3; 2.63 ± 0.48) was low.

### 3.3. Gut Microbial Composition

The overall gut microbial composition of all fish samples at the phylum level displayed diverse assemblages of microbes consisting mainly of Proteobacteria (79% of the total sequences), Firmicutes (16%) and Bacteroidetes (3%), with other phyla (Actinobacteria, Cyanobacteria, Saccharibacteria, Fusobacteria, Acidobacteria, Tenericutes, Deinococcus-Thermus and Verucomicrobia) accounting for less than 2% of the total sequences.

When the taxonomic composition of each fish species at the phylum level was assessed, Proteobacteria was found to be the dominant phylum in the four coral fish species with other codominant phyla. *C. urodeta*, *L. kasmira*, and *G. aurolineatus* exhibited substantial proportions of Proteobacteria (82%, 81%, and 63%, respectively) and Firmicutes (16%, 17%, and 32%, respectively), while *O. unifasciatus* was dominated by Proteobacteria (89%) and Bacteroidetes (9%) (Figure 3). Little variation was observed in phylum-level taxonomic composition among replicates.

The taxonomic composition at the genus level was markedly more variable both between and within each fish species (Figure 4). *Escherichia-Shigella* was the dominant genus in *O. unifasciatus* and *G. aurolineatus* (43% and 53%, respectively). The most abundant genus in *C. urodeta* and *L. kasmira* was *Psychrobacter* (25% and 50%, respectively), while *C. urodeta* was characterized by the codominance of *Escherichia-Shigella* (15%), an unclassified genus of Clostridiaceae (12%), *Vibrio* (20%), and *Photobacterium* (14%). A phylogenetic tree was generated based on the 191 clustered OTUs and illustrated the top four phyla and top 30 genera in the four fish species (Figure 5). There was great intraspecies variability at the individual level, and the composition of the gut microbes in each fish species at the genus level may therefore be of doubt in some instances (Figure 4). For instance, the abundance of *Vibrio* in the three replicates of *C. urodelus* ranged from 0.01% to 46.45%. An exception to this pattern was observed for *G. aurolineatus*, whose replicates were consistently dominated by *Escherichia-Shigella* (43–73%).

### 3.4. Core and Unique Gut Microbial Community in Four Coral Reef Fishes from the South China Sea

The core gut microbes (the proportion of the microbial community shared between samples) were investigated for each species as a result of the observed high intraspecies variability. The proportions of the 24 shared OTUs among the replicates of each species were remarkably low, accounting for 8–13% of the total microbial OTUs (Figure 6). However, they accounted for a large fraction of the total sequences (68–84%). The proportions of shared sequences in the replicates of *O. unifasciatus*, *C. urodeta*, *L. kasmira*, and *G. aurolineatus* were 64–86%, 77–89%, 72–91% and 61–87%, respectively.

Many of the shared OTUs among the replicates for a species came from the abundant taxa in that species. For instance, the four fish species mainly shared *Escherichia-Shigella*, *Psychrobacter*, *Vibrio* and unclassified genera within phylum Clostridia. Other shared OTUs in the four fish species belonged to *Acinetobacter*, *Burkholderia-Paraburkholderia*, *Bradyrhizobium*, *Propionibacterium*, and an unclassified genus of Cyanobacteria. In addition, a small proportion of sequences belonging to *Ralstonia*, *Arthrobacter*, *Propionibacterium*, *Tepidimonas*, *Methylobacterium*, *Pelomonas*, *Pseudomonas*, and *Acinetobacter* were recovered from the four fishes.

The differences in the microbial communities of the four studied fishes were tested with via ANOSIM (R = 0.352, *p* = 0.018), showing no statistically significant differences in the microbial community among the four fish species (Figure 7). After further analysis, it was found that there were still some unique microbial OTUs recovered in different fish species (Figure 6 and Table 2). For example, *Alkanindiges*, *Lysobacter*, *Olsenella*, *Paenibacillus*, *Porphyromonas*, *Rubritalea*, *Solibacillus*, and *Sphingopyxis* were only recovered from the gut of *O. unifasciatus*, and 13 genera, including *Carnobacterium*, *Desulfovibrio*, *Dolosigranulum*, *Endozoicomonas*, *Helcococcus*, *Massilia*, *Streptomyces*, and five unclassified genera, were only detected in the gut of *C. urodeta*. In addition, the unique microbial community of *L. kasmira* consisted of *Anaerotruncus*, *Clostridium*, *Deinococcus*, *Faecalibacterium*, *Hymenobacter*, *Macellibacteroides*, *Nesterenkonia*, *Nocardiopsis*, *Peptococcus*, *Rheinheimera*, *Roseomonas*, and *Staphylococcus*, while six genera, *Abiotrophia*, *Bosea*, *Gemella*, *Haliscomenobacter*, *Leptotrichia*, and *Veillonella*, and an unclassified genus within *Erysipelotrichaceae* constituted the unique gut microbial community of *G. aurolineatus*.

## 4. Discussion

### 4.1. New Insights into the Gut Microbial Communities of Coral Reef Fishes

There is an apparent lack of research on the intestinal microflora of coral reef fishes, especially among carnivorous fish species from the South China Sea. This is the first investigation of the gut microbial communities of four carnivorous coral fish species (*O. unifasciatus*, *C. urodeta*, *L. kasmira*, and *G. aurolineatus*) from the South China Sea using high-throughput Illumina sequencing. In this study, an intriguing finding was that a total of 20 microbial genera, including *g_11-24*, *Abiotrophia*, *Aquabacterium*, *BD1-7_clade*, *Bosea*, *Chryseomicrobium*, *Dolosigranulum*, *Erysipelotrichaceae_UCG-006*, *Family_XIII_AD3011_group*, *Haliscomenobacter*, *Leptotrichia*, *Leucobacter*, *Macellibacteroides*, *Massilia*, *Nocardiopsis*, *Olsenella*, *Peptococcus*, *Roseomonas*, *Ruminococcaceae_NK4A214_group*, *Ruminococcaceae_UCG-014*, and *Tepidimonas*, were recovered for the first time from the guts of marine fishes (Table 3), which increased the number of gut microbial genera known from marine fishes. Among these 20 microbial genera, eight genera (indicated by asterisks in Table 3) are newly reported as gut microbes of fishes. The remaining 12 genera were previously reported in the gut microflora of freshwater fishes such as *Ctenopharyngodon idellus* [18], *Fundulus heteroclitus* [19], *Oncorhynchus mykiss* [20,21], *Danio rerio* [22], *Carassius auratus* [23], *Carassius gibelio* [24], and *Coregonus baicalensis* [25]. With the rapid development of modern molecular biotechnology, an increasing number of novel microbial communities and functions have recently been detected in the guts of fishes by using new techniques and methods [7,8,9]. Our results suggested that carnivorous coral reef fishes from the South China Sea harbor many previously unknown components of the gut microbial community.

### 4.2. Richness and Diversity Estimates of Gut Microbiota in Coral Reef Fishes

The richness and diversity estimates of the gut microbiomes of the four coral reef fishes varied substantially (Chao1, 35–98; Shannon, 1.98–4.08) with an average Shannon’s index of 2.76. These values were compared to estimates from other carnivorous coral reef fish species including *Centropristis striata* (120 ± 80.7; 3.1 ± 2.76), *Scomberomorus cavalla* (143 ± 42.4; 1.79 ± 0.05), *Caranx hippos* (160 ± 47.7; 4.24 ± 2.59), *Sphyraena barracuda* (28.2 ± 0.99; 0.69 ± 0.87), *Carcharhinus brevipinna* (107 ± 102; 2.31 ± 0.11) and *Naso hexacanthus* 178 ± 18; 2.85 ± 0.53) [11,26]. However, conclusions obtained through different pipelines can be inconsistent, so the published diversity estimates are used for comparison as qualitative values [27]. The alpha diversity detected in this study suggested variation in richness between replicates in the guts of some coral reef fishes. This may be due to the complexity of the coral skeleton and reef structure in the coral ecosystem and the nutritional structure of the coral ecosystem.

### 4.3. Comparison of Gut Microbial Community in Different Coral Reef Fish Species

Several previous studies comparing the microbiota associated with coral reef fish species collected from the Gulf of Mexico and Southern Great Barrier Reef have shown a marked specificity between fish species and microbiota [28,29,30,31]. However, the results of ANOSIM conducted in this study indicated high similarity between the four fish species, probably because the four captured fishes mainly feed on small fishes and crustaceans, according to which the four fish species are classified as carnivores. This finding may demonstrate a certain correlation between the fish gut microflora and the diet of the fish.

In a recent study, Miyake et al. revealed that many members of Firmicutes, especially in the *Epulopiscium* genus, constituted the most abundant microbial community in the guts of seven herbivorous coral reef fishes [11]. In contrast, Proteobacteria was the dominant phylum (63–89%) in the four coral reef fish species included in the present study. *C. urodeta*, *L. kasmira* and *G. aurolineatus* exhibited codominance of Firmicutes (> 15%). In *O. unifasciatus*, Firmicutes accounted for less than 1%, but Bacteroidetes exhibited a relatively high abundance (9%), including *Chryseobacterium*, *Empedobacter* and *Flavobacterium*. It has been reported that *O. unifasciatus* and *C. urodeta* feed mainly on fishes (83% and 68%, respectively) and crustaceans (14% and 32%, respectively); *L. kasmira* also feeds on fishes (42%) and crustaceans (31%) but consumes a variety of algae as well; and *G. aurolineatus* feeds mainly on crabs and gastropods, also consuming small fishes occasionally [32,33,34]. Among the four fish, *O. unifasciatus* and *C. urodeta* have a similar diet, with small fishes accounting for a large proportion of the diet, but there was a significant difference in the codominant phyla of the guts of these fish species, perhaps because *C. urodeta* is distantly related to *O. unifasciatus* phylogenetically (Figure 1). Our results suggested that the diet might play a more important role in shaping the gut microbiota of carnivorous coral reef fishes than the species of fish.

### 4.4. Core and Unique Microbial Community in Different Coral Reef Fish Species

A total of 24 OTUs affiliated with 15 genera were shared between the four coral reef fishes (core microbial community). Although the proportions of shared OTUs in each fish species were relatively low, accounting for 8–13% of the total microbial OTUs, the shared sequences accounted for a large fraction of the total sequences (68–84%) in each fish species. A similar result was found in a recent study by Miyake et al. [11]. Among the identified core gut microbiota, *Psychrobacter*, *Escherichia-Shigella*, and *Vibrio* were the most dominant genera in the four coral reef fishes. *Psychrobacter* is a widespread and evolutionarily successful microbial genus that can be recovered in high abundance from the guts of different marine fishes, such as *Salvelinus alpinus* [35] and *Gadus morhua* [36]. *Psychrobacter* has been reported to improve autochthonous microbial diversity along the GI tract and to improve feed utilization and innate immunity in *Epinephelus coioides* [37,38]. As conditional pathogens, *Escherichia-Shigella* and *Vibrio* are frequently detected in many healthy Actinopterygii fish species [39] and Atlantic salmon parr [40].

Although unique gut microbial genera accounted for a small proportion of the total sequences, there were many such genera in each coral reef fish species (Table 2). Interestingly, one to four of the genera among the unique gut microbiota in each fish species constituted new reports for marine fishes (Table 2). Most of the observed unique gut microbial genera can be frequently recovered from marine fishes. For example, *Endozoicomonas* spp. are often isolated from cardinalfish (Apogonidae) and damselfish (Pomacentridae) [41], and *Clostridium* spp. and *Staphylococcus* spp. are frequently recovered in pinfish (*Lagodon rhomboids*) [26,36].

## 5. Conclusions

In summary, the gut microbiota of four carnivorous coral reef fishes, *O. unifasciatus*, *C. urodeta*, *L. kasmira*, and *G. aurolineatus*, from the South China Sea were successfully characterized by high-throughput Illumina sequencing. Proteobacteria along with Firmicutes and Bacteroidetes constituted 98% of the gut microbiota of the four coral reef fishes from the South China Sea, and a total of 20 gut microbial genera recovered in this study represented new reports for marine fishes. ANOSIM indicated that the four carnivorous coral reef fishes harbored a similar microbial community, suggesting that diet type (carnivorous) might play a more important role in shaping the gut microbiota in coral reef fishes than the fish species. In addition, *Psychrobacter*, *Escherichia-Shigella*, and *Vibrio* constituted the core microbial community of the four carnivorous coral reef fishes. Although the unique gut microbial genera accounted for a small proportion of the total sequences, there were many such genera in each coral reef fish species, including several genera (*Endozoicomonas*, *Clostridium* and *Staphylococcus*) that are frequently found in marine fishes and 11 new reports of gut microbes in marine fishes.

With the rapid development of modern molecular technology, an increasing number of gut microbiota in different marine organisms have been investigated using high-throughput sequencing methods. However, the relatively high analysis costs do not allow a large number of samples, and so it must be based on only a small number of samples, which may mean that the relatively limited data would have been different if supported by a higher sample base. Unfortunately, this is still the limit of this study. Therefore, more than three replicated samples should be included in the experiment in order to obtain more accurate data and results. Certainly, triplicate analysis on the same subject will be acceptable but not encouraged.

## Figures and Tables

**Figure 1 microorganisms-08-00426-f001:**
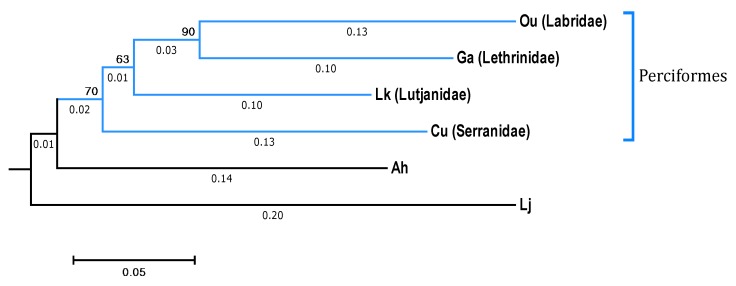
Neighbor-Joining phylogenetic tree of the cytochrome c oxidase subunit I (COI) gene of the four coral reef fishes from the South China Sea with the maximum composite likelihood and bootstrap method (Q = 1000) was rooted by Ah (*Acanthogobius hasta*) and Lj (*Lampetra japonicum*) and constructed with mega 6.0. The clone sequences of the fish were downloaded from NCBI and the locus_tags were KC353468.1, NC_006131.1, MK567523.1, MK658630.1, FJ416614.1, MK566923.1. Ou: *O. unifasciatus*, Cu: *C. urodeta*, Lk: *L. kasmira*, Ga: *G. aurolineatus*.

**Figure 2 microorganisms-08-00426-f002:**
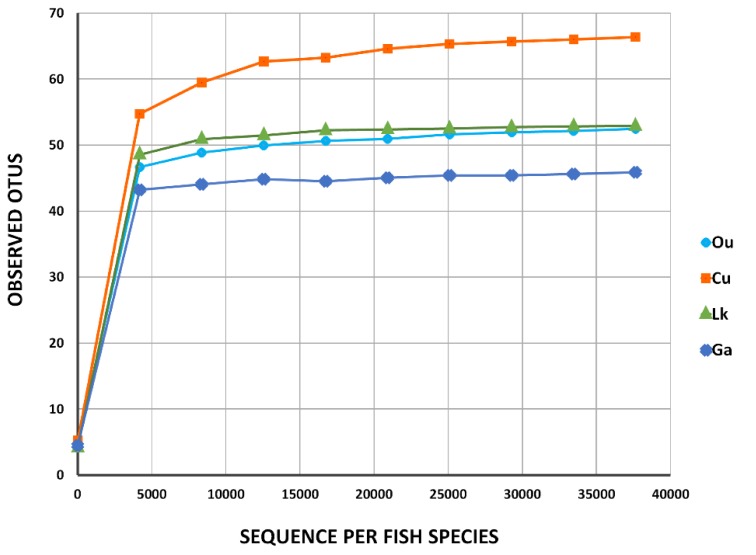
Rarefaction curves of four coral reef fishes from the South China Sea. Ou: *O. unifasciatus*, Cu: *C. urodeta*, Lk: *L. kasmira*, Ga: *G. aurolineatus*.

**Figure 3 microorganisms-08-00426-f003:**
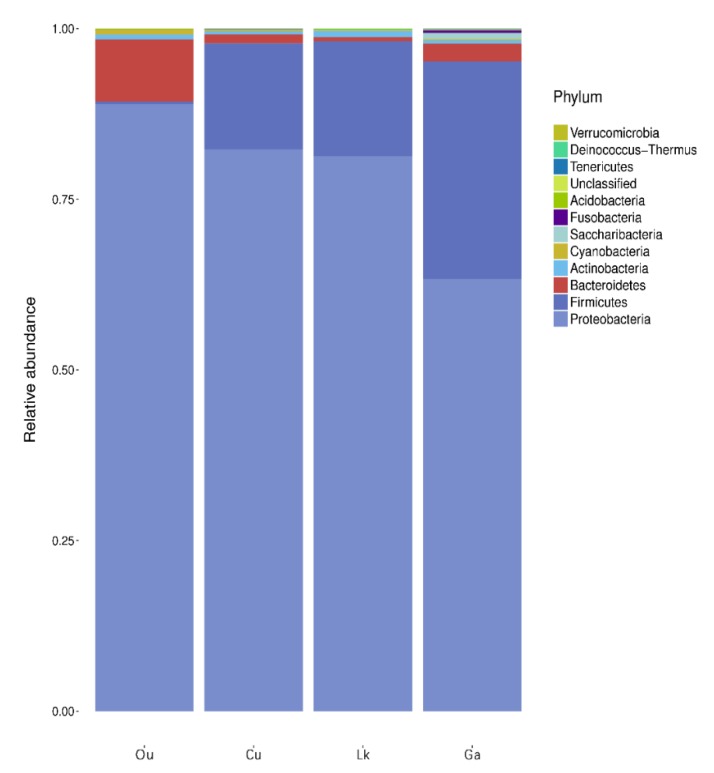
Taxonomic composition of gut microbes in four coral reef fishes at phylum level. Ou: *O. unifasciatus*, Cu: *C. urodeta*, Lk: *L. kasmira*, Ga: *G. aurolineatus*.

**Figure 4 microorganisms-08-00426-f004:**
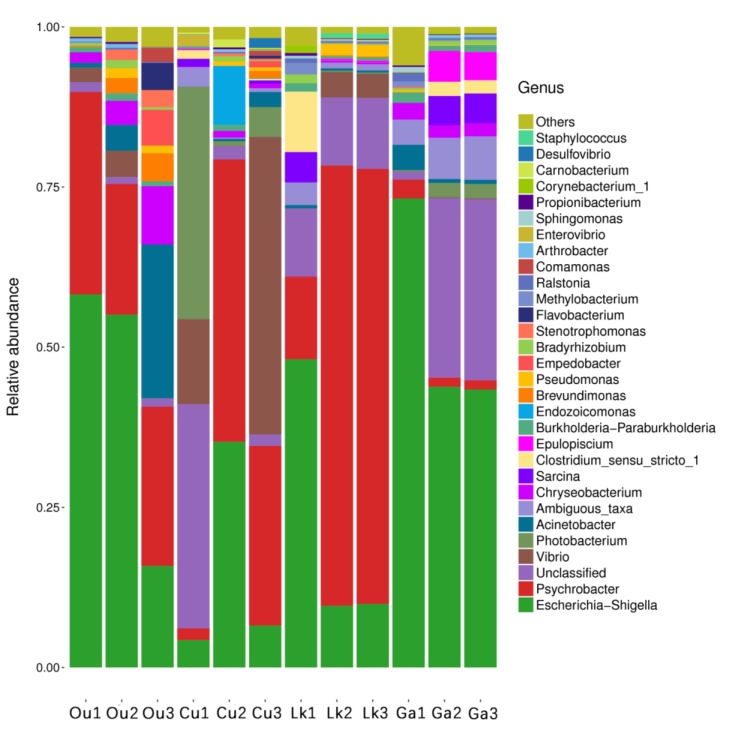
Interspecies gut microbial communities classified at the genus level. Ou: *O. unifasciatus*, Cu: *C. urodeta*, Lk: *L. kasmira*, Ga: *G. aurolineatus*.

**Figure 5 microorganisms-08-00426-f005:**
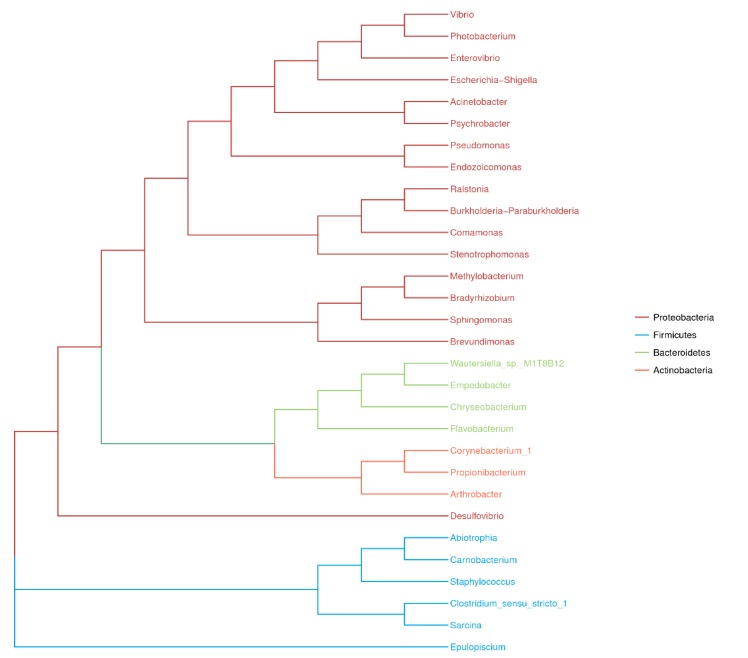
Phylogenic tree of the top 30 most abundant microbial OTUs recovered in the four fish species from the South China Sea revealed by high-throughput Illumina sequencing

**Figure 6 microorganisms-08-00426-f006:**
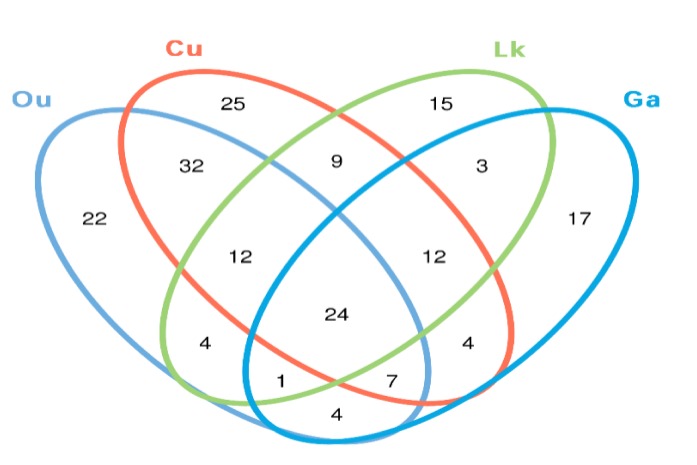
Venn diagram illustrates overlap of OTUs in gut microbiota among the four fish species. Ou: *O. unifasciatus*, Cu: *C. urodeta*, Lk: *L. kasmira*, Ga: *G. aurolineatus*.

**Figure 7 microorganisms-08-00426-f007:**
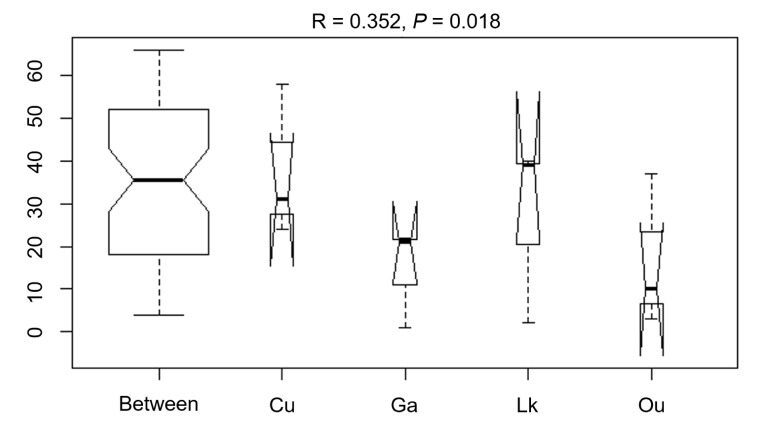
ANOSIM comparison (R statistics) of the four coral reef fishes from the South China Sea. Ou: *O. unifasciatus*, Cu: *C. urodeta*, Lk: *L. kasmira*, Ga: *G. aurolineatus*.

**Table 1 microorganisms-08-00426-t001:** Comparison of microbial sequences, the estimated operational taxonomic units (OTUs) richness, Shannon’s indexes of the microbial 16S V4 sequences for clustering at 97% sequences similarity from the sequencing analysis.

Species of Coral Fishes	Sequence Read	Avg Len (bp)	OTUs	Chao1	Shannon	Simpson	ACE
*Oxycheilinus unifasciatus*	53195	461	106	54	2.89	0.71	54.45
*Cephalopholis urodeta*	69114	459	125	67	2.96	0.77	67.81
*Lutjanus kasmira*	52213	458	80	53	2.55	0.65	53.84
*Gnathodentex aurolineatus*	51425	454	72	46	2.63	0.65	47.75

**Table 2 microorganisms-08-00426-t002:** Unique gut microbial community at genus level in four different coral reef fishes from the South China Sea.

Fish Species	Unique Microbial Community
*Oxycheilinus unifasciatus*	*Alkanindiges*, *Lysobacter*, *Olsenella* *, *Paenibacillus*, *Porphyromonas*, *Rubritalea*, *Solibacillus*, *Sphingopyxis*
*Cephalopholis urodeta*	*BD1-7_clade*, *Carnobacterium*, *Desulfovibrio*, *Dolosigranulum* *, *Endozoicomonas*, *Family_XIII_AD3011_group*, *Helcococcus*, *Massilia* *, *Ruminococcaceae_NK4A214_group*, *Ruminococcaceae_UCG-014*, *Streptomyces*, *and uncultured Legionellales bacterium*
*Lutjanus kasmira*	*Anaerotruncus*, *Clostridium*, *Deinococcus*, *Faecalibacterium*, *Hymenobacter*, *Macellibacteroides* *, *Nesterenkonia*, *Nocardiopsis* *, *Peptococcus* *, *Rheinheimera*, *Roseomonas* *, *Staphylococcus*
*Gnathodentex aurolineatus*	*Abiotrophia* *, *Bosea* *, *Erysipelotrichaceae_UCG-006*, *Gemella*, *Haliscomenobacter* *, *Leptotrichia* *, *Veillonella*

The genera marked by asterisk (*) are new reports for gut microbes of fishes.

**Table 3 microorganisms-08-00426-t003:** Summary of new reports for gut microbiota in marine fishes.

Microbial Genera	Source Fishes				Isolated /Clone	Reference
	Ou	Cu	Lk	Ga		
*g_11-24* *	√				clone	in this study
*Abiotrophia*				√	clone	in this study
*Aquabacterium*	√		√		clone	in this study
*BD1-7_clade* *		√			clone	in this study
*Bosea*				√	clone	in this study
*Chryseomicrobium* *	√		√	√	clone	in this study
*Dolosigranulum* *		√			clone	in this study
*Erysipelotrichaceae_UCG-006* *				√	clone	in this study
*Family_XIII_AD3011_group* *		√			clone	in this study
*Haliscomenobacter*				√	clone	in this study
*Leptotrichia*				√	clone	in this study
*Macellibacteroides*			√		clone	in this study
*Massilia*		√			clone	in this study
*Nocardiopsis*			√		clone	in this study
*Olsenella*	√				clone	in this study
*Peptococcus*			√		clone	in this study
*Roseomonas*			√		clone	in this study
*Ruminococcaceae_NK4A214_group* *		√			clone	in this study
*Ruminococcaceae_UCG-014* *		√			clone	in this study
*Tepidimonas*	√	√	√	√	clone	in this study

The genera marked by asterisk (*) are new reports for gut microbes of fishes, Ou: O. unifasciatus, Cu: C. urodeta, Lk: L. kasmira, Ga: G. aurolineatus.

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
