# Peer review of "Deep Insights into Gut Microbiota in Four Carnivorous Coral Reef Fishes from the South China Sea"

_microorganisms, 2020, doi:10.3390/microorganisms8030426_

Round 1

Reviewer 1 Report

The manuscript microorganisms-743642 describes the composition of the gut microbiota by sequencing with Illumina of four carnivorous fish species of coral reef from southern China. The study is well described and clearly supported in all its aspects, exhaustively describing the procedure performed. Unfortunately works like this, where the most advanced technology is used with relatively high analysis costs that do not allow a large number of samples, it must be based only on a small number of these; the data would have been different if supported by a higher sample base; unfortunately it is still the limit of these studies. However, the authors performed well the triplicate of analysis on the same subject. Despite everything, the results obtained are extremely important and support the conclusions that the authors provide. The cited bibliography is well selected and valid for this manuscript. The authors are asked to include in the part of the materials and methods relating to the sample collection, to better explain how the intestine samples were treated before freezing at -80 ° C. Also on page 10 line 16, the name of the species must be mentioned in lowercase (Oncorhynchus mykiss). For all these considerations and for the importance of the data obtained made available, in my opinion the paper does not require any particular changes or clarifications and can be accepted for publication after minor revision.

Author Response

  1. The authors are asked to include in the part of the materials and methods relating to the sample collection, to better explain how the intestine samples were treated before freezing at -80 ° C.

Response: the material and methods relating to the sample collection were revised, which was as follows.

These coral reef fishes were collected using hook and line, and kept in steriled sea water. After anaesthetization with 60 mg/L tricaine methanesulfonate, each fish was transported on ice to the laboratory as soon as possible [11]. Under aseptic conditions, the external surface of the fish was cleaned with 75% ethanol to avoid the contamination by the surface microbes on the fish. After opened the ventral surface using steriled scissors, the whole gut of each fish was aseptically removed and collected, and at least three replicates per sample were collected and assembled as one gut sample. A total of 12 gut samples from the four coral reef fishes were collected in frozen tubes and stored at -80 ℃ until DNA extraction.

  1. Also on page 10 line 16, the name of the species must be mentioned in lowercase (Oncorhynchus mykiss).

Response: Changed “Oncorhynchus Mykiss” to “Oncorhynchus mykiss”.

Reviewer 2 Report

The manuscript titled “Deep insights into gut microbiota in four carnivorous coral reef fishes from the South China Sea” investigates the gut microbial diversity among fish to provide baseline data for wild marine fish. This is an important field of study as the fish-associated symbiotic gut microbiota plays a crucial role in nutritional provision and metabolic homeostasis. The manuscript has been written very well. The figures support the overall conclusions of the study. I would recommend accepting the manuscript in its current form.

Author Response

  1. The manuscript titled “Deep insights into gut microbiota in four carnivorous coral reef fishes from the South China Sea” investigates the gut microbial diversity among fish to provide baseline data for wild marine fish. This is an important field of study as the fish-associated symbiotic gut microbiota plays a crucial role in nutritional provision and metabolic homeostasis. The manuscript has been written very well. The figures support the overall conclusions of the study. I would recommend accepting the manuscript in its current form.

Response: The authors are grateful to the reviewer for his/her time, and valuable and positive comments.